# Mixed Land Use Levels in Rural Settlements and Their Influencing Factors: A Case Study of Pingba Village in Chongqing, China

**DOI:** 10.3390/ijerph19105845

**Published:** 2022-05-11

**Authors:** Hongji Chen, Kangchuan Su, Lixian Peng, Guohua Bi, Lulu Zhou, Qingyuan Yang

**Affiliations:** 1Chongqing Jinfo Mountain Karst Ecosystem National Observation and Research Station, School of Geographical Sciences, Southwest University, Chongqing 400715, China; chj001292@email.swu.edu.cn (H.C.); biguohua@email.swu.edu.cn (G.B.); zllzxnlcg@email.swu.edu.cn (L.Z.); 2Institute of Green and Low-Carbon Development, Southwest University, Chongqing 400715, China; 3College of State Governance, Southwest University, Chongqing 400715, China; sukangchuan@163.com; 4School of Geography and Tourism, Shaanxi Normal University, Xi’an 710119, China; plixian@163.com

**Keywords:** mixed land use, rural settlement, influence factor, village planning, Pingba Village

## Abstract

Mixed land use provides an important means of promoting the intensive and efficient use of land resources and stimulating endogenous development power in rural areas. This paper selected Pingba Village in Chongqing as the research area; the land use status data and the social and economic data on rural settlements in the study area for 2021 were obtained through field visits and interviews. Moreover, the land use types in the rural settlements were subdivided according to the principle of dominant function. Based on these subdivisions, a land mixed-use measurement system for rural settlements was constructed to analyze their levels of mixed land use. Furthermore, the influences of natural environmental, social, economic and other factors on mixed land use were comprehensively explored. The results showed that, (1) the mixed land use of rural settlements in the study area was at a medium level and showed significant spatial variability, and rural settlements in the high, medium and low mixed land use index zones accounted for 12.5%, 35% and 52.5% of the total, respectively. (2) The differences in the natural environment determined the level of mixed land use and the basic pattern of its spatial differentiation. Social and economic factors, such as resident population and average household income, were key impact factors. Rural tourism resources, homestead agglomeration policies and other factors had important impacts on the level of mixed land use. In conclusion, the research suggests that mixed land use is an important way to boost rural revitalization. In the future, village planning could introduce the concept of mixed land use to improve the efficiency of land use, optimize the land use structure according to local conditions and promote the integrated development of rural primary, secondary and tertiary industries. In addition, it is necessary to scientifically and rationally guide rural settlements to agglomerate appropriately to improve the utilization efficiency of land resources and public service resources.

## 1. Introduction

The mixed use of two or more land uses in a certain space is called mixed land use (MLU) [1], a concept resulting from critical reflections by urban planning scholars on the disadvantages of urban land functional zoning in the middle of the 20th century [2]. It has received extensive attention as an effective way to solve urban land use problems and improve intensive land utilization. With the development of global urbanization, the rough use of rural land and rural decline have become international problems, and promoting rural revitalization and intensive use of rural land is a common challenge facing the world [3,4]. In China, with the implementation of the urban–rural integration development and rural revitalization strategy, new industries and businesses have emerged in rural areas, such as at rural–urban fringes and the eastern coastal region of China; thus, the land use structure of rural settlements has evolved in diversity and complexity, leading to multifunctional rural settlements with residence, operation, leisure and public services [5]. In 2019, the Chinese government issued the “Opinions on Coordinating and Promoting Village Planning”, calling for the formulation and implementation of village planning, overall planning for the layout of land use in villages and improvement in the living environment and proposing controls for the scale of residential land in villages in accordance with the principle of economical and intensive land use. Furthermore, the Central Committee of the Communist Party of China (CPC) and the State Council issued the “Opinions on the Key Work of Comprehensively Promoting Rural Revitalization” in 2022, which called for continuing to promote the integrated development of the primary, secondary and tertiary industries in rural areas and accelerating the implementation of land use policies to guarantee and regulate the integrated development of the primary, secondary and tertiary industries in rural areas. As an important way to promote the intensive and efficient use of land resources, MLUs can effectively alleviate the shortage of land for rural industrial development. For implementing rural revitalization strategies, studying MLU in rural settlements for the compilation and implementation of village planning, the sustainable development of rural areas and the intensive use of rural land resources is of great significance.

MLU has always been a hot research field for urban planning scholars and geographers and mainly focuses on the following aspects: (1) The connotation and characteristics of MLUs were first proposed by Jacobs in his book The Death and Life of Great American Cities [6], which received widespread attention. Rowley and Alan [7] were the first to comprehensively introduce the concept of mixed development and stated that the spatial physical form of MLUs should be reflected in urban structure and the environment. Moreover, some scholars have thought that MLU is relative to the concept of single land use and refers to specific areas adjacent to plots with different land use types [8,9,10,11]. Furthermore, with continuing research, the understanding of MLU has gradually deepened. It is now believed that MLU not only includes the quantity and scale of land types but also includes many characteristics of land functional layout, spatial form, etc. [2]. (2) The measurement system and research method, as based on multiple understandings of its connotations, includes four types of MLU measurement systems and methods. First, based on diversity, the MLU level can be measured by the quantitative and scale dimensions of land use such as the diversity index [12], Simpson diversity index [13] and entropy index [14]. Second, based on accessibility, the MLU level can be measured by the spatial structure dimension of land use such as the pedestrian index [15] and proximity [16]. Third, based on compatibility, the MLU level can be measured by the promiscuity [17] and compatibility [18] methods, which measure the functional relationship dimension of land use. Fourth, based on the comprehensive dimension, the MLU level can be measured by a comprehensive measurement system constructed from the quantitative scale, spatial structure and functional relationship of land use [19,20]. (3) The social and economic effects of MLU have been found in studies to have an important impact on urban sustainable development capacity [21], urban housing prices [22,23,24], regional development vitality [25], urban transportation [26,27] and residents’ social activities and health [28,29]. In addition, some scholars have begun to pay attention to the phenomenon of MLU in rural settlements. Some scholars have explored the transformation of land use in rural settlements and the multi-functionalization of rural settlement land from the perspective of land multifunctionality [30,31,32,33]. Zhu et al. [34] and Wu et al. [35] discussed the mixed characteristics and spatial form of rural settlement land from the perspective of the work–live community. Moreover, Zhang et al. [1,36] proposed a research framework for MLU in rural settlements and carried out an empirical analysis of Yao Village in Tianjin. They deemed that socioeconomic transformation and the self-adjustment of residents’ production and lifestyle were the main driving factors of MLU, and their research provided a very valuable reference for the development of this paper.

In summary, the research methods of MLUs have evolved from a single perspective to multidimensional comprehensive measurement, and the research content has evolved from the connotation identification of MLUs and construction of mixing level measurement systems to the external effects and driving mechanisms of MLUs. In addition, scholars have begun to pay attention to the phenomenon of MLU in rural areas. In conclusion, the existing research results are fruitful and provide guidance and references for our research. However, there are also some shortcomings: (1) At present, studies on MLU are mainly concentrated in urban areas, and there are relatively few studies on rural settlements. Therefore, empirical studies from different regions and perspectives are much needed. (2) Compared with cities, rural settlements exhibit substantial differences in land use scales and spatial forms, especially in mountainous and hilly areas; these areas have observable spatial characteristics of “large dispersion, small agglomeration and small scale”; thus, the methods used to measure the spatial structure of urban MLUs cannot be applied directly to the study of rural settlements, necessitating the exploration of a new measurement method. (3) Most scholars have used land compatibility to measure the function of the relationship between different land types. Compatibility refers to a state in which two or more land use types coexist without significant negative effects [17,18]; compatibility can reflect the functional relationship between various land types but cannot reveal the extent of interactions between different land types and coupling relationships. Fortunately, coupled coordination models can effectively compensate for this deficiency.

Pingba Village, located in the hinterland of the Three Gorges Reservoir area, was once one of the poverty-stricken villages in Chongqing. In recent years, with state support, emerging industries, such as rural tourism and homestays, have been vigorously developed, and land use patterns and structures have become increasingly diversified, demonstrating clear characteristics of mixed land use. Hence, this paper took Pingba Village as the research area, analyzed the level and spatial differentiation of mixed land use in rural settlements by constructing a measurement system for MLU and discusses the impacts of the natural environment, social economy, tourism resources, local policies and other factors on MLU in order to provide a reference for optimizing settlement land layout and promoting intensive and efficient land use in the study area or other villages in the Three Gorges Reservoir area.

## 2. Materials and Methods

### 2.1. Study Area

Due to the fact of its geographical location, resource endowment, industrial foundation and other conditions, some villages in Chongqing have developed distinct industrial characteristics of “one village one product” and “one township one industry”. Of these, villages with rural tourism, rural homestays and new efficient industries at their core are provided as representative models of Chongqing and mountainous villages in Southwest China. Intensely developed rural emerging industries have led the MLU phenomenon to be more common in these villages; therefore, they are appropriate representatives for this research.

Pingba Village is one of the demonstration villages of “one village one product” in Chongqing, one of the first batch of beautiful and livable villages in China and one of the beautiful leisure villages in China. Since 2018, the village has diversified its economic structure by rapidly developing rural tourism, homestays and new efficient rural industries. According to the field survey, there is a quantity of commercial and public service land, and the rural settlement land clearly demonstrates typical features of MLU.

Rural settlements have various types of living situations where rural residents gather to live and work [32]. Moreover, the heterogeneity of the physical geographical environment makes the spatial form of rural settlements differ greatly among different regions. In contrast with rural settlements in plain areas, rural settlements in mountainous and hilly areas show the spatial characteristics of “large dispersion and small agglomeration”; therefore, from market towns to single homesteads, multiple forms of rural settlements in hilly and mountainous areas exist. According to the survey, there are 40 village settlements within the administrative area of Pingba Village (Figure 1), which has the spatial pattern characteristics of rural settlements in typical mountainous and hilly areas.

Located in the southeast of Zhongyi Township in Chongqing, Pingba Village is adjacent to Quanxing Village in the east, Yanjing Village in the west, Huaxi Village in the north, and Shasha Town in the south. It is 48 km from Shizhu County and 9 km from Zhongyi Xiangchang town. The altitude is between 800 and 1680 m. Pingba Village has seven groups of 1440 villagers under its jurisdiction. By the end of 2019, its collective disposable economic income was 360,000 yuan with a per capita disposable income reaching 12,800 yuan.

### 2.2. Data Sources

The data used in this paper mainly include geospatial data and socioeconomic data of the study area (Table 1). In order to achieve the purpose of the study, this paper adopted the integrated survey method of “field investigations + interviews with farmers”, which involves both the mapping of physical information and the acquisition of socioeconomic information of farmers’ households. In addition, we obtained data on basic household information, residential base utilization, household employment and household economic income of farm households in Pingba Village through farm interviews. The household interviews were conducted in the form of a census, targeting all permanent households in Pingba Village.

### 2.3. Research Methods

#### 2.3.1. Land Use Classification of Rural Settlements in the Study Area

As a territorial space for the production, life, social and cultural activities of rural residents, rural settlements are land use complexes that include a variety of land types such as rural homesteads, public service lands and commercial service lands [37,38]. However, most scholars view rural settlements as a whole, which creates difficulties for deeply revealing the diverse land use structural and functional characteristics within rural settlements and which offers insufficient support for the refined management of rural settlements. Therefore, through field investigations and from the perspective of multifunctional land, this paper further subdivided the internal land use of rural settlements in the study area into 11 types of single-function land types and 4 types of multifunctional mixed-land types according to the principle of dominant function. Single-function land types refer to land with only one functional use in a plot. On the contrary, multifunctional mixed-land types refer to land with two or more functional uses at the same time (Table 2).

#### 2.3.2. Connotation and Measurement System of MLU in Rural Settlements

(1)Connotation of MLU in rural settlements

Many scholars understand the connotation of MLU in urban settlements as well as the research results about the work–live community of rural settlements. This paper argues that the MLU of rural settlements refers to the coexistence of two or more different land use types in rural settlements, which is driven by multiple factors such as the diversified demands of rural residents and foreign consumers for rural land use, rural socioeconomic transformation and the implementation of national rural policies and strategies. Moreover, its connotation emphasizes the diversity of land use types, the proximity of spatial structure and the coordination of functional relations. The diversity of land use also includes the staggered use of multiple blocks in plane space and the compound use of different functional spaces in stereoscopic space (Figure 2).

(2)The Measurement System of MLU Levels

The measurement of MLU is the basis of studying MLU in rural settlements. Based on existing studies [1,2] and considering the availability of data, this paper constructed a measurement system of MLU in rural settlements from the three dimensions of quantitative scale, spatial structure and functional relationship (Table 3) to measure the level of MLU in rural settlements in the study area.

#### 2.3.3. Measurement Methods and Models

(1)Simpson diversity index

The diversity of land use is a basic indicator used to measure the degree of mixed land use [41]. The Simpson diversity index, an ecological measure of biodiversity, was used to measure the extent of MLU of rural settlements in the study area in 2021 on a planar spatial basis using the following formula:(1)$$SIM=1-\sum_{i=1}^{n}\frac{S_{i}^{2}}{S^{2}}\;$$
where SIM is the degree of MLU in the plane space; $S_{i}$ is the area of the i-th land use type of the colony; S is the total area of the colony land. The range of SIM was [0, 1], and the larger the value, the higher the degree of mixed land use in the horizontal direction.

(2)Vertical mixing index

Vertical mixed land use (VMLU) refers to the mixture of two and more functional spaces in the vertical direction on the same plot [8]. Based on the reference to existing studies [1], the vertical mixing index was used to measure the degree of mixed land use in the vertical direction in rural settlements in the study area in 2021, and the formula is:(2)$$V=\frac{S_{f}}{S_{0}\left\{\sum_{i}^{N}\left[\left(f_{i}\right)\left(\frac{s_{i}}{s_{f}}\right)\right]\right\}}\left(i=1,2,3\ldots ,n\right)$$
where V is the vertical mixing degree index; $S_{f}$ is the total area of all statistical functional spaces in the settlement; $S_{0}\;$ is the total area of the plot; $\frac{s_{i}}{s_{f}}$ represents the functional intensity of the plot;$\;s_{i}$ is the area of the ith functional space in the settlement;$\;f_{i}$ is the number of layers of the i-th functional space; $\left\{\sum_{i}^{N}\left[\left(f_{i}\right)\left(\frac{s_{i}}{s_{f}}\right)\right]\right\}$ denotes the average number of floors of N land use types in the settlement.

(3)Compactness model

Compactness is an important index to reflect the spatial form and structure of settlements as well as the overall efficiency of the spatial layout of geographical things [40]. It is generally believed that a circular layout has the highest overall efficiency; thus, the circular area was taken as the standard unit of measurement, and the compactness value of circular ground objects was 1, while the compactness of other ground objects was less than 1.
(3)$$K=\frac{2\sqrt{\pi A}}{P}\;$$
where K represents the compactness of the rural settlement; A represents the village area; P represents the contour perimeter of the rural settlement. The larger K is, the more compact the spatial pattern of the settlement and the higher the overall efficiency of land use.

(4)Coupling coordination degree model

Coupling refers to the degree of influence and the interaction between two or more systems or elements [8]; therefore, our study used the coupling degree to analyze the functional relationship between different land classes in rural settlements to reflect the level of MLU in rural settlements in depth. Moreover, based on reference to existing literature [42,43,44], the land use function was divided into 4 primary functional classes: production function, living function, ecological function and cultural function. In addition, there were differences in the functional strength of different land classes; thus, this paper introduces the concept of strong/semi/weak functionality [45], subdivided the primary functional class into 12 secondary functional classes, and then assigned values to the functions of different land classes in rural settlements (Table 4). On this basis, the coupling degree model was used to measure the coupling relationship between different functions of land use in rural settlements. However, when the traditional coupling degree model calculates the coupling degree of multiple systems, if there exists a geographical system function with a value of 0, the coupling degree of the whole system is all 0, which is not in line with the reality of system interrelationships. Fortunately, Zhang Yu et al. [46] adopted the coefficient of variation to modify the coupling coordination degree model which solved the deficiency of the traditional coupling degree model. The formula is as follows:(4)$$C=\sqrt{2-\frac{4(P_{i}^{2}+R_{i}^{2}+E_{i}^{2}+F_{i}^{2})}{(P_{i}+R_{i}+E_{i}+F_{i}){}^{2}}}\;$$
where $P_{i},R_{i},E_{i}\;\mathrm{and}\;F_{i}$ denote the rating values of the production function, living function, ecological function and cultural function of the i-th land use type, respectively. However, the coupling degree model can reflect only the strength of the coupling effect of each system and cannot avoid the phenomenon of low-level high coupling or high-level low coupling [47]. To reflect the level of coordinated development, the study introduced a coordination degree model that considered the strength of the interaction between the four functions and could reflect the level of development of each. The formula is as follows:(5)$$D={\left(C\cdot T\right)}^{\frac{1}{2}};\;T=\alpha P_{i}+\beta R_{i}+\gamma E_{i}+\delta F_{i}\;$$
where D is the coupling degree; T is the coordination degree;α,β,γ,δ are the system function coefficients. Based on the principle of balanced and synergistic development of regional functions, the functional roles of each system in the study area are equally important; therefore, each functional coefficient was one-quarter.

(5)Composite index of MLU

MLU contains three characteristics of land use: diversity, spatial proximity and functional coordination; therefore, its calculation formula is:(6)M=αSIM+βC+γK+δD 
where M is the composite index of the MLU; α,β,γ,δ are the system function coefficients. Moreover, the three-dimensional characteristics of the quantitative scale, spatial structure and functional relationship of MLU should be equal weights; thus, each coefficient was one-quarter.

#### 2.3.4. Multiple Linear Regression Model

Multiple linear regression analysis describes the linear dependence between a dependent variable and two or more independent variables, and multiple independent variables are used to jointly predict or estimate the trend of the dependent variable [48]. Therefore, to clarify the socioeconomic factors affecting the spatial variation in MLU in rural settlements, the multiple linear regression model was selected for regression analysis of the possible influencing factor variables with the following model equation:(7)$$Y=\beta_{0}+\beta_{1}X_{1}+\beta_{2}X_{2}+\ldots +\beta_{n}X_{n}+\varepsilon \;$$
where Y is the dependent variable, representing the difference in the levels of MLU in rural settlements; $\beta_{0}$ is a constant term; $X_{i}\;$(i = 1, 2, …, n) is a series of socioeconomic factors that may differ with the levels of MLU. Furthermore, after fieldwork and screening them one at a time, indicators were selected from three dimensions: settlement population size, settlement farm household characteristic and homestead use (Table 5).

## 3. Results

### 3.1. Level and Spatial Characteristics of MLU in Rural Settlements

Using the comprehensive measurement system of MLU in rural settlements, the degree of MLU of rural settlements in the study area was analyzed, and the level of MLU of rural settlement land in the study area (Table 6) and its spatial distribution map (Figure 3) were obtained. Using the natural breakpoint method, the MLU index of rural settlements in Pingba Village in 2021 was divided into high-value zones (0.6, 0.74], medium-value zones (0.45, 0.6] and low-value zones [0, 0.45]. In general, the mean value of the MLU index in rural settlements in Pingba Village was 0.47, which was at the medium level. The high-value, medium-value and low-value zones of the MLU index accounted for 12.5%, 35% and 52.5%, respectively.

The level of MLU in rural settlements showed significant regional differences. Specifically, the medium- and high-value zones of the MLU index were mainly concentrated in gullies and valleys with low topography and convenient transportation. In particular, rural settlement No. 2, 8 and 14 all had an MLU index of 0.7 or more. In contrast, the areas with low values on the MLU index were mainly located in high terrain and inaccessible high mountain areas, with the lowest value of 0.24 in colony No. 32.

### 3.2. Influencing Factors of MLU

(1)Natural factors

The natural geographical environment determined the basic pattern of the MLU level in rural settlements. Among them, natural factors, such as elevation, slope and river, had a large influence on the MLU level. A three-dimensional map of the spatial distribution of the mixed land use levels and natural factors in rural settlements (Figure 4), based on QGIS 3.22, reveals the spatial coupling between them. Figure 3 shows that the spatial distribution pattern of the MLU levels of rural settlements in Pingba Village had obvious topographic directionality and river directionality. The median and high-value areas of the MLU index were mainly concentrated in the valley areas with low terrain, where there were enough water sources, convenient transportation, abundant arable land resources, large-scale settlements and various land use methods; this is also prime area for developing rural tourism. The development of rural tourism, rural homestays and other industries in Pingba Village demonstrate the obvious nature of the MLU of rural settlement in this area.

(2)Socioeconomic factors

The significance of the multiple linear regression analysis was 0.000, which is less than 0.05, indicating a significant linear relationship between these six socioeconomic indicators on the level of mixed land use. The degree of influence of each variable (Table 7) was as follows: $X_{4}$ > $X_{6}$ > $X_{1}$ > $X_{3}$ > $X_{2}$ > $X_{5}$. The significance of the three indicators—average annual income of settlement households ($X_{4}$), settlement building structure ($X_{6}$) and settlement resident population ($X_{1}$)—were all less than 0.05 and significant, indicating that they had a strong influence on the level of mixed land use in settlements. In addition, the significance of the number of households carrying out business activities ($X_{2}$) and the number of people with an education above high school ($X_{3}$) was less than 0.1, indicating that the effects of these two indicators on the spatial differences in the level of mixed land use in rural settlements were not significant and had some explanatory effect. The significance of the number of layers of settlement buildings ($X_{5}$) was not strong. As the subject of land use, humans are the core of rural regional systems: therefore, the population size of rural settlements is the key factor affecting the level of mixed land use. The larger the population size of rural settlements, the greater the intensity of land use, the more diverse the use of land and the higher the degree of the mixed-use of land. Moreover, the higher the average annual income of rural settlements, the stronger the economic power, the more diverse the demand for land use types and the higher the ability to invest in business activities and improve farmhouses and surrounding habitat.

(3)Tourism resource factors

There was a correlation in the study area between the MLU level in rural settlements and rural tourism resources. Through the introduction of social capital, Pingba Village has created Dawan Homestay, Xiangjiaba Homestay and other tourism experience projects, driving the development of more than 40 rural tourism reception households in the village and forming a tourism line that integrates education and research, leisure and tourism and farming experience. The rapid development of the rural tourism industry has led to the diversification of land use in rural settlements. Due to the demonstration effect and radiation of the Dawan Homestay, farmhouses, kiosks, picking gardens and other businesses have emerged in the surrounding settlements (Figure 5); additionally, a variety of land types, such as parking lots, cultural plazas and green belts, have developed, which constantly promote MLU in rural settlements, improve land use efficiency and enhance rural development vitality. Settlement No. 14, as the central area of Pingba Village, was at a high level of plot diversity and land use functional coupling coordination after village planning and land consolidation.

(4)Policy factors

Local policies profoundly affect the MLU level in rural settlements. As a special settlement form scientifically planned and centrally built by government departments, the population scale, land use scale, spatial structure and infrastructure of rural settlements far exceed those formed in the natural state and are at a high level in terms of quantity and scale of land use, spatial structure and functional relationship. Rural settlement No. 2 in the study area was used as a resettlement site for poverty alleviation in Pingba Village. Its population scale reached more than 200 people, and its interior covered a variety of land types, such as residential land, yard land, green land, road land and public service land, with a high MLU index of 0.72. In summary, as an important driving force, guiding force and binding force of rural social and economic development, local policies have a profound impact on the evolution of land use patterns, structures and functions of rural settlements and, thus, have an important impact on the level of MLU.

## 4. Discussion

### 4.1. Innovation and Deficiency

As a complex of production, life and social interaction of rural residents, rural settlements carry the diversified needs of rural residents and urban consumers; thus, the phenomenon of MLU in rural settlements has a certain degree of universality. This paper further subdivided the land use within rural settlements of Pingba Village in Chongqing into 11 single-function land classes and four mixed-land classes according to the dominant function principle, which better revealed the diverse land use structure and functional characteristics within rural settlements. Settlement compactness and coupling coordination methods were introduced to analyze the characteristics of MLU in rural settlements for the two dimensions of spatial structure and functional relationship. Compared with the existing research, this paper incorporated the coupling coordination degree of land functions and the mixing degree of vertical space into the measurement system of MLU, which is innovative to a certain extent and can provide certain insights for the preparation of village planning and research on mixed land use in the study area and other rural areas. However, most studies use the indicator of land compatibility to measure the relationship between different land uses, and then land compatibility cannot reveal the degree of the interaction and coupling relationship between different land types; therefore, this indicator was not used in this paper. Rather, it reflected this aspect through coupling coordination of land use. MLU is an inevitable trend of socioeconomic transformation in rural areas and is universal, and the phenomenon of MLU is common in both China and other countries in the world. In the context of global rural decline and rough land use, promoting mixed land use in rural areas is undoubtedly significant for improving rural vitality and residents’ quality of life. This paper took Pingba Village in China as an example to explore the MLU in rural settlements, and the results of the study provide a Chinese case reference for international rural revitalization and rural governance.

However, this paper also has some shortcomings: (1) Due to the availability of data and the difficulty of field investigation, this paper selected only Pingba Village with mountainous characteristics as the research object, which could not fully and objectively reflect the characteristics of MLU of rural settlements of various geomorphic types. Furthermore, the interaction of multiple factors, such as the natural environment, resource endowment, agricultural production, the phenomenon of migration and socioeconomic development level, has resulted in significant spatial heterogeneity in the MLU of rural settlements. Therefore, in the future, rural settlements in different geographical regions, such as plains, plateaus and hills, should be selected to conduct comparative studies and explore the general rules of MLUs in rural settlements in order to develop suggestions for promoting rural revitalization and improving the intensive and efficient use of rural land. (2) MLU is the dynamic evolution of the interaction between people and the environment in rural areas and is the result of the interactions between multiple interests in the rural area system; its essential connotations have social and temporal attributes. Therefore, building a more comprehensive measurement system considering the social relationship dimension of rural settlements, exploring the spatial and temporal evolution characteristics of MLUs and their driving mechanisms and optimizing and reconfiguring the land use space of rural settlements from the perspective of MLUs are key areas that need to be studied in the future.

### 4.2. Policy Implications

#### 4.2.1. Introduce the Concept of MLU into Village Planning

In recent years, China has begun to build a territorial space planning system to strengthen space planning and governance and is currently in the initial stage of exploration. Village planning, as the basis and detailed planning of the territorial space planning system, is the content that every village must complete so as to serve as the legal basis for establishing territorial space development and protection activities, implementing territorial space use control and carrying out various construction activities in rural areas, and it plays an important role in guiding and guaranteeing the promotion of rural revitalization. With limited construction land resources in rural areas, improving the efficiency of rural land use and providing land security for rural revitalization projects in the future is a major challenge for the majority of rural areas. MLU is an important way to promote the intensive and efficient use of land resources. In the future, the concept of MLU can be introduced into the process of village planning and implementation to moderately optimize the land use layout of rural settlements, improve the efficiency of land use in rural areas and promote the intensive use of rural land resources and high-quality development.

#### 4.2.2. Optimize the Land Use Structure according to Local Conditions and Promote the Integrated Development of Primary, Secondary and Tertiary Industries in the Countryside

MLU can provide land security for industrial development. Industrial development can drive the adjustment and optimization of land use and structure, and the two complement each other. In the future, the natural environment and resource endowment of villages can lead to the development of rural industries, and the integration of agriculture with processing, tourism, culture and other industries can be vigorously promoted. Moreover, ecological resources can be used to develop ecological tourism, leisure, recreation and other industries to stimulate the development of a village’s collective economy while protecting the ecological environment. In addition, it is possible to revitalize land resources and activate rural development vitality by supporting, encouraging and guiding farmers to use their own farmhouses to carry out new industries and new businesses such as rural homestays, farm caravans and e-commerce.

#### 4.2.3. It Is Necessary to Scientifically Guide the Appropriate and Reasonable Clustering of Rural Settlements

In fact, the MLU level can objectively reflect the size of the resident population and the living environment of rural settlements and other realities. The results of this paper show that the areas with low levels of MLU were generally rural settlements with high terrain, inconvenient transportation, poor infrastructure and small settlement land and population sizes. In contrast, the policy-led residential settlements showed high levels of MLU. Rural settlements in the mountainous areas of Southwest China were characterized by “large scattered, small clusters, small scale and sporadic distribution”. As part of rural revitalization, the goal of “five connections” (i.e., water, electricity, roads, gas and network) for each household has to be realized; this realization brings substantial investments in infrastructure. Therefore, to save national public infrastructure investment and reduce repetitive waste, centralized settlements can be appropriately carried out. To respect farmers’ wishes and needs, rural settlements with high terrain, small-scale and poor living environments are guided toward larger rural settlements or centralized resettlement areas where infrastructure support services can be shared and MLU promoted.

## 5. Conclusions

This paper constructed a measurement system of MLU for rural settlements and used qualitative and quantitative methods to measure the MLU levels of rural settlements in the study area and to analyze the spatial differences in these MLU levels and their influencing factors. The main conclusions are as follows:(1)MLU is an inevitable trend of social and economic transformation in rural areas. As an important part of rural regional systems, land use structures are spatial representations of human–environment interactions in rural areas. MLU is discussed from the perspective of the human–environment interaction within rural settlements. It is beneficial to expand and enrich the scientific connotation of rural area system theory and the multifunction theory of land use. In practice, guidance for the compilation and implementation of village planning, the adjustment and optimization of rural settlement land structure and the intensive use of rural land resources can be provided;(2)The degree of MLU in rural settlements in the study area was at a medium level, and there were significant differences in spatial patterns. Specifically, the percentages of rural settlements in the high-, medium- and low-value zones of MLU were 12.5%, 35% and 52.5%, respectively. The medium- and high-value zones of MLU were mainly located in the valley areas with low terrain and convenient transportation, while the low-value zones of MLU were mainly located in the high mountain areas with high terrain and inconvenient transportation;(3)The coupling interaction of multiple factors, such as the natural environment, resource endowment and local policy, profoundly affected the MLU level and its spatial differentiation pattern in the rural settlements. The differences in terrain, slope and other natural environments in the study area determined the level of MLU and the basic pattern of spatial differences. Social and economic factors, such as resident population and average household income, were key factors affecting the level of MLU. Tourism resources and homestead agglomeration policies can improve MLU levels;(4)In the future, the concept of MLU can be introduced into village planning to appropriately optimize the layout of rural settlements and promote the intensive use and high-quality development of rural land resources. Moreover, industrial development can drive the adjustment and optimization of land use patterns and structures, allowing the study area and other villages in the Three Gorges Reservoir area to develop rural industries according to local conditions and promote the integrated development of primary, secondary and tertiary industries. In addition, the utilization efficiency of land resources and public service resources can be improved by guiding the moderate and reasonable agglomeration of rural settlements, promoting MLU and optimizing the layout of land use.

## Figures and Tables

**Figure 1 ijerph-19-05845-f001:**
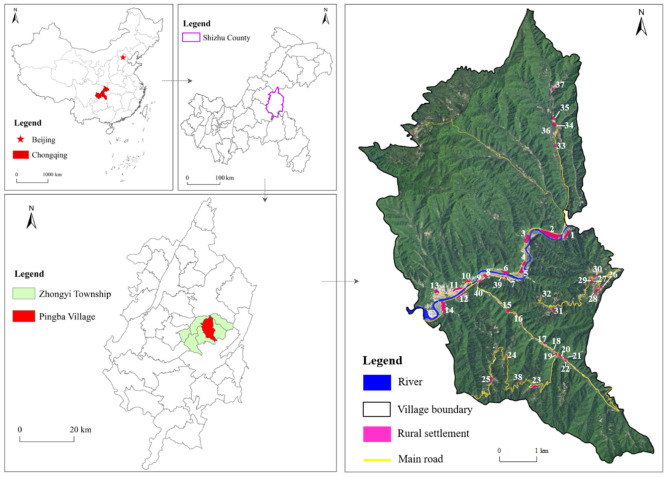
The geographic location of the study area and the numbering map of rural settlements. The white numbers in the figure is the number of rural settlements.

**Figure 2 ijerph-19-05845-f002:**
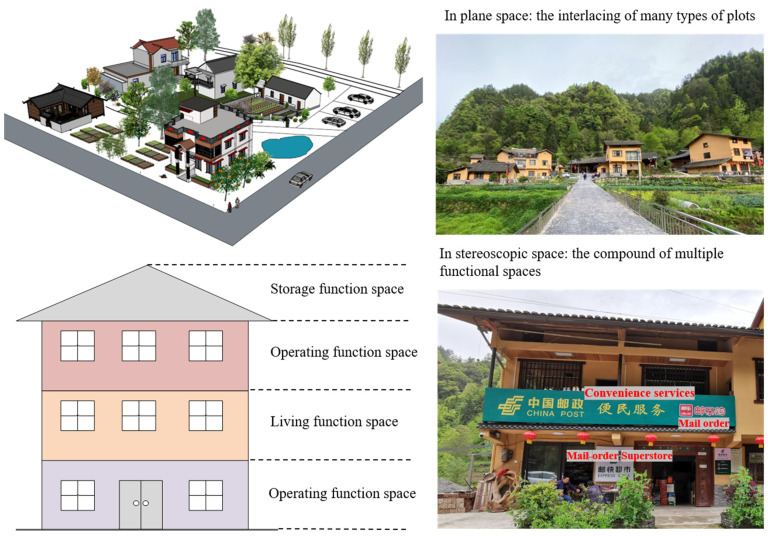
The conceptual model and real landscape structure of MLUs in rural settlements.

**Figure 3 ijerph-19-05845-f003:**
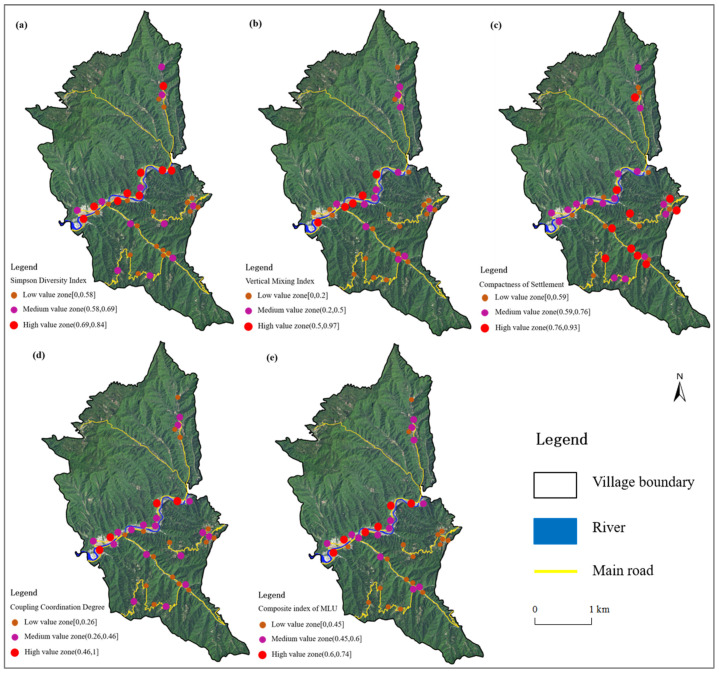
Spatial characteristics of the MLU of rural settlements in the study area in 2021. (**a**) the diversity index; (**b**) the vertical mixing index; (**c**) the compactness of settlement; (**d**) the coupling coordination degree; (**e**) the composite index of MLU.

**Figure 4 ijerph-19-05845-f004:**
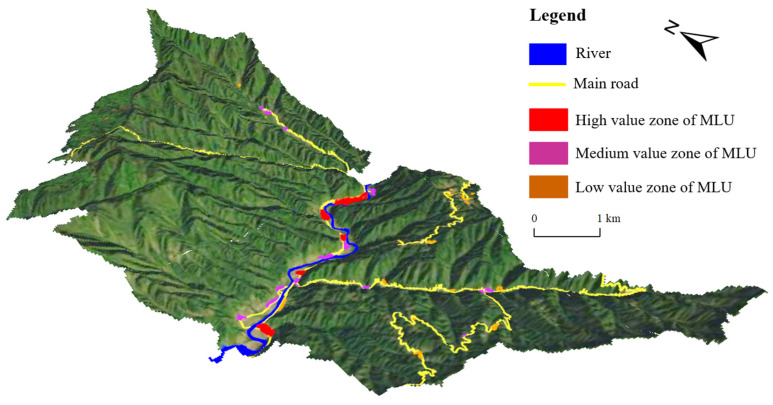
The coupling relationship between the horizontal spatial characteristics of MLUs in rural settlements and the natural environment in the study area in 2021.

**Figure 5 ijerph-19-05845-f005:**
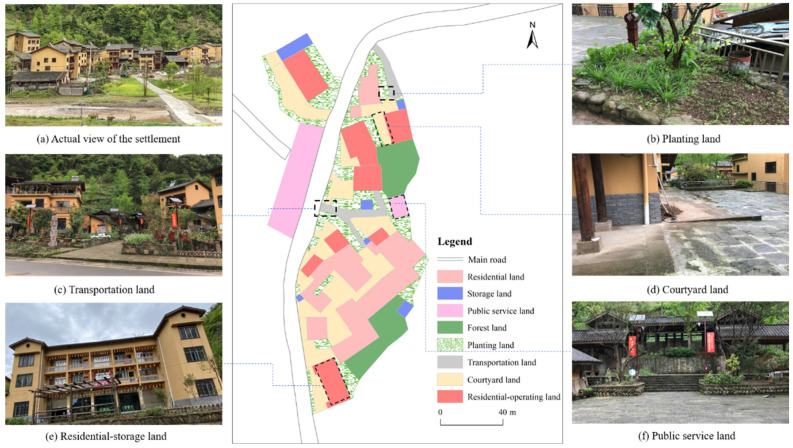
Schematic diagram of the structure of the MLU in settlement No. 14 (Xiangjiaba Homestay).

**Table 1 ijerph-19-05845-t001:** Data types and sources.

Data Types		Explanation	Data Source
Geospatial data	Land use data	Using remote sensing images as the base map, the land use map within the rural settlements in the study area was mapped by field survey.	Field investigations
	DEM	Digital elevation model; its resolution was 30 m.	The China Geospatial Data Cloud (http://www.gscloud.cn/home (accessed on 2 April 2021))
	Administrative zoning	Included data such as the township boundaries of Shizhu County and the. administrative boundaries of Pingba Village.	Chongqing Shizhu County Planning and Natural Resources Bureau
	Remote sensing image data	Based on ArcGIS10.2 and QGIS3.22 software cropping.	The China National Platform for Common Geospatial Information Services (https://www.tianditu.gov.cn/ (accessed on 2 April 2021))
	Settlement buildings	Included the structure, number of floors and internal functional space of the colony building.	Field investigations
Socioeconomic data	Number of people in the settlement	Included the household and resident population of the settlement.	The villagers’ committee of Pingba Village
	Agricultural and economic statistics report	Included data on land area, village collective income and village industrial development.	The villagers’ committee of Pingba Village
	Educational level of the farmers	The higher the education level of the farmers, the higher their knowledge and ability to accept new things.	Interviews with farmers
	Employment of the farmers	Included the employment status, nature of employment and income of farm households.	Interviews with farmers
	Annual household income of farm households	Refers to the total annual economic income of the household.	Interviews with farmers

**Table 2 ijerph-19-05845-t002:** Land use classification of rural settlements in the study area in 2021.

Land Types	Connotation
Residential land	Land for buildings that support the daily life, rest and residence of rural residents.
Planting land	Land used for planting vegetables, corn and other crops to meet the production and living needs of farmers.
Livestock land	Land used for breeding chickens, ducks, geese, pigs and other livestock.
Courtyard land	Open space for residents’ production, life, leisure and entertainment (such as the “Baba Dance”) and other functions.
Operating land	Land used for business activities such as homestays, farmhouses and canteens.
Storage land	Land used for storing farm tools, firewood, agricultural products and other daily sundries.
Transportation land	Roads used for transportation within the settlement area, excluding township roads and above.
Forestland	Forests that are closely related to the production and life of the residents within the settlement range.
Green land	Land for planting plants with greening and ornamental functions such as planting shrubs and flowers.
Industrial production land	Land for industrial productive activities such as breweries and edible mushroom processing plants.
Public service land	Land with public services such as village service centers, village post stations, and parking lots.
Residential—operating land	Plots of land with residential and business functions such as villagers using their own farmhouses to operate a farmhouse or snack shop.
Residential—storage land	Land on the same plot with dual functions of living and storage such as the storage of farm tools or other household objects in the basement or top floor of the farmhouse.
Forest—livestock land	Inner forest land of the settlement used for the farming of chickens, ducks, geese and other poultry.
Residential—public service —operating land	Land on the same plot with multiple functions of living, operating and public services, such as villagers using their own houses to operate farm entertainment or small stores as well as postal stations or express services.

**Table 3 ijerph-19-05845-t003:** Measurement system of MLU in rural settlements.

Measure Dimension		Measure Basis	Connotation	Method
Number and size	Plane Space	Diversity of land use	Drawing on the ecological biodiversity index [39], the area ratio of different land types is calculated.	Simpson diversity index
	Stereo space		Drawing on the concept of floor area ratio, it is the average number of floors per unit of functional space [1].	Vertical mixing index
Spatial structure		Compactness of rural settlements	Reflects the overall efficiency of the spatial layout of land use and indirectly reflects the degree of mixing [40].	Compactness model
Functional relationship		Coupling coordination	Reflects the degree of interaction and influence between different land types. The greater the degree of coupling coordination, the higher the level of mixed land use.	Coupling coordination degree model

**Table 4 ijerph-19-05845-t004:** The production–living–ecological–cultural function assignment table of rural settlement land in the study area in 2021.

Primary Function Class	Secondary Function Classes and Assignment	Land Category
Production function	Strong production function (9)	Planting land, livestock land, operating land, industrial production land, residential—operating land and residential—public service—operating land.
	Semi-production function (5)	Courtyard land, transportation land and forest—livestock land
	Weak production function (3)	Residential land, storage land, green land and residential—storage land
Living function	Strong living function (9)	Residential land, residential—operating land, residential—public service—operating land and residential—storage land
	Semi-living function (5)	Planting land, public service land, storage land, forest—livestock land, operating land, courtyard land and livestock land
	Weak living function (3)	Industrial production land and green land
Ecological function	Strong ecological function (9)	Forest land and forest—livestock land
	Semi-ecological function (5)	Green land
	Weak ecological function (3)	Planting land
Cultural function	Strong cultural function (9)	Residential land, residential—operating land, residential—public service—operating land and residential—storage land
	Semi-cultural function (5)	Courtyard land and Public service land
	Weak cultural function (3)	Operating land and industrial production land

**Table 5 ijerph-19-05845-t005:** Socioeconomic influencing factor variables and selection basis of the levels of MLU.

Variable Category	Independent Variable	Variable Selection Basis
Population size	Resident population of rural settlements ($X_{1}$)	People are the main body of land use, and the resident population of the settlements is the key factor in the type, structure and function of land use.
Farmer characteristics	Number of households carrying out business activities ($X_{2}$)	Business activities, such as country houses and farmhouses, can promote the diversification of land use functions.
	Education level above high school ($X_{3}$)	The higher the education level, the stronger the education level and the ability to accept new things.
	Average annual income of settlement households ($X_{4}$)	The income of farming households is an important indicator of the economic strength of the settlement, and the higher the income, the higher the ability to transform land use practices.
Architectural features	The total number of floors of settlement buildings ($X_{5}$)	The number of building floors affects the size and type of functional space in the vertical direction.
	Settlement building structure ($X_{6}$)	The building structure of a colony affects the number of floors and its internal environment and has a significant impact on the conduct of operational activities. It was quantified according to the level of quality: reinforced concrete structure = 9; brick and concrete = 7; brick and wood = 5; civil/stone and wood = 3; all wood = 1.

**Table 6 ijerph-19-05845-t006:** Level of MLU for rural settlements in the study area in 2021.

Settlement Number	Total Area of Land (m^2^)	Number of Land Types	Total Number of Floors	MLU Level	Settlement Number	Total Area of Land (m^2^)	Number of Land Types	Total Number of Floors	MLU Level
1	11,967.94	7	38	0.54	21	1618.38	5	2	0.33
2	44,849.64	7	84	0.72	22	933.90	5	2	0.44
3	12,325.92	9	58	0.66	23	5804.95	5	6	0.4
4	5179.39	5	17	0.6	24	742.53	3	6	0.38
5	8434.08	8	16	0.57	25	3351.47	5	3	0.38
6	7151.07	8	18	0.59	26	3351.07	6	9	0.41
7	1406.07	4	11	0.51	27	2073.70	3	6	0.41
8	4370.29	7	18	0.71	28	2471.08	4	6	0.42
9	2190.04	3	18	0.51	29	3459.69	5	6	0.43
10	4783.27	5	12	0.51	30	1595.74	4	4	0.44
11	9150.38	7	12	0.5	31	3505.86	6	2	0.42
12	4692.58	4	3	0.35	32	225.30	2	2	0.24
13	6205.29	5	2	0.45	33	1640.68	3	8	0.51
14	14,784.26	9	44	0.74	34	5113.85	5	6	0.51
15	4025.41	5	12	0.5	35	3467.04	5	14	0.51
16	881.97	3	4	0.43	36	376.60	2	4	0.37
17	1446.64	4	5	0.43	37	1935.19	4	4	0.41
18	452.31	2	2	0.35	38	899.09	4	3	0.49
19	2325.08	4	8	0.53	39	892.62	4	2	0.37
20	3422.65	4	6	0.53	40	448.72	2	2	0.45

**Table 7 ijerph-19-05845-t007:** Regression coefficient table of MLU levels and socioeconomic factors.

Variable Category	Independent Variable	Standardized Coefficient	Significance	VIF
Population size	Resident population of rural settlements $\left(X_{1}\right)$	0.603	0.013	9.600
Farmer characteristics	Number of households carrying out business activities $\left(X_{2}\right)$	0.087	0.092	5.996
	Education level above high school $\left(X_{3}\right)$	0.530	0.087	12.007
	Average annual income of settlement households $\left(X_{4}\right)$	0.476	0.000	1.625
Architectural features	Total number of floors of settlement buildings $\left(X_{5}\right)$	0.307	0.327	9.176
	Settlement building structure $\left(X_{6}\right)$	0.269	0.011	2.331

VIF (variance inflation factor) was used to measure the covariance problem of the independent variables—the larger the value, the more severe the covariance.

## Data Availability

Not applicable.

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
