# Peer review of "Mixed Land Use Levels in Rural Settlements and Their Influencing Factors: A Case Study of Pingba Village in Chongqing, China"

_ijerph, 2022, doi:10.3390/ijerph19105845_

Round 1

Reviewer 1 Report

The innovative contribution of this paper seems to be in two dimensions: analysis of mixed land use (MLU) in rural (rather than urban) contexts; extension of quantitative MLU measures to rural contexts.

 “Field investigations” are mentioned in several places in the article. Yet they are not included among the data sources. We need more information on these field investigations in the section on data sources.

As far as I understand the methodology, the quantitative MLU index is developed in detail as a four-dimensional measure with equal weights. For some reason, the authors do not use the term “equal weights” and instead give an incomprehensible explanation on lines 265-266. In this sentence, the terms “balanced and synergetic” led me to suspect equal weighting in formula (6). If this is indeed so, the explanation should be reworded; if not, a clear explanation for the weights in (6) should be provided.  

Compare your composite MLU index to similar MLU indexes in the literature. Especially tell us which of your components are included (or not) in alternative indexes and which components used in other sources are omitted from your index.

References [1, 2] appear to be the key methodological references underpinning the study. Unfortunately, they are both in Chinese. Some explanation of the importance of [1, 2] to the present study should be provided.

Another technical reference related to formula (4) concerns the coupling degree correction model [40] is also in Chinese. Some explanation of this model in English would be welcome.

Since most of the existing literature deals with urban MLU, it is important to try and identify the few references (preferably in English) that nevertheless discuss MLU in the rural context.

On line 289, we read that the MLU index “was divided into high, medium and low-value zones”. The authors need to tell us the range (both theoretical and empirical) of the MLU index values and what cutoff points are used to divide the continuous spectrum into low, medium, and high. This information is hidden in the map legends in Figure 3, but we need to see it explicitly in the body of the text.

I do not see the purpose of table 5 with detailed values of the MLU index and its four components for all 40 villages studied. In my opinion, it is much better to present a table with summary statistics instead of a full listing of the individual values.

Nor do I see any justification for the detailed ANOVA results in table 6. It is probably sufficient to say that the regression is statistically significant (in the sense that at least one of the coefficients is nonzero).

In table 7, explain (in a footnote) what VIF stands for.

Make sure to use consistent terminology: always use “residential-operating” (instead of “residential-operate”) and “service-operating” (instead of “service-operate”).

The adjective “breeding” (as in “breeding land” and “forest-breeding land”) sounds awkward. Normally you would characterize this type of land use as “grazing land”. If you think that “grazing” is inappropriate because you want to include also chickens, ducks, geese, and pigs, perhaps you should use the term “livestock land”.

In table 1, clearly separate between the single-use types and the mixed-use types.

On line 202, LMU is probably MLU mistyped.

On lines 204-205, the phrase "quantitative scale-spatial structure-functional relationship" (referring to table 2) is totally unclear. Please reword in a meaningful way.

Reviewer 2 Report

Title: Mixed land use levels in rural settlements and their influencing

factors: A case study of Pingba Village in Chongqing, China

Journal: International Journal of Environmental Research and Public Health

Comments of the reviewer

  1. The article deals with the important issue of the correct directions of rural development in the aspect of mixed land use (MLU) as a phenomenon that can contribute to sustainable development of rural areas with the simultaneous intensification of land use.
  2. The research is interesting and presents a different approach than the most common two-dimensional analyzes. The research methods used should be considered sufficient and applied correctly. Perhaps even the authors have tried to capture too many aspects analyzed in this one study. Which is not a suggestion for changes in this regard. However, the issue is so complicated and interesting that I see a chance for future extended research, especially the relationship between the observed dependencies (MLU) and the profile of agricultural production in a given area, the level of income, or the phenomenon of migration.
  3. What is the state of China's spatial planning with regard to rural development? are local spatial development plans obligatory for every village? Are there any limitations as to the function (type of use) of the buildings being constructed?
  4. I have not found information about the data sources and methods used to obtain the primary information sets.
  5. Figure 1. I would like to see the layout of property boundaries (plots) in the analyzed area, in particular in the area of ​​the village center. The process of rural development is largely dependent on the spatial arrangement of plots, in many countries this is both a limitation and a challenge to spatial planning processes in built-up agricultural areas. It is then necessary to take actions of the nature of land consolidation or other types of reorganization of the plot boundary system.
  6. The biggest disadvantage of the present form of the article, which requires improvement, is the need for a broader approach to the discussed issue from an international perspective. The authors mainly use (with a few exceptions) the results of research related to China. However, at the stage of introduction and discussion, the background of the problem and the discussion of the obtained results should also be placed in a broader perspective. The issues raised are universal and also occur in other regions of the world.
